# Role of Nanocrystallites of Al-Based Glasses and H_2_O_2_ in Degradation Azo Dyes

**DOI:** 10.3390/ma14010039

**Published:** 2020-12-24

**Authors:** Qi Chen, Zhicheng Yan, Hao Zhang, KiBuem Kim, Weimin Wang

**Affiliations:** 1Key Laboratory for Liquid-Solid Structural Evolution and Processing of Materials (Ministry of Education), School of Materials Science and Engineering, Shandong University, Jinan 250061, China; caesar@mail.sdu.edu.cn (Q.C.); rengaryzc@outlook.com (Z.Y.); zhanghao_0611@163.com (H.Z.); 2Department of Nanotechnology and Advanced Materials Engineering, Sejong University, 98 Gunja-dong, Gwangjin-gu, Seoul 143-747, Korea; kbkim@seijong.ac.kr

**Keywords:** Al-based glasses, methyl orange dye, nano-scale crystallites, H_2_O_2_

## Abstract

Al-based metallic glasses have a special atomic structure and should have a unique degradation ability in azo dye solutions. The Al_88_Ni_9_Y_3_ (Y3), Al_85_Ni_9_Y_6_ (Y6) and Al_82_Ni_9_Y_9_ (Y9) glassy ribbons are melt spun and used in degrading methyl orange (MO) azo dye solution with adding H_2_O_2_. With increasing *c*_Y_, the as-spun ribbons have an increasing GFA (glass formability) and gradually decreased the degradation rate of MO solution. TEM (transmission electron microscopy) results show that the Y3 ribbon has nano-scale crystallites, which may form the channels to transport elements to the surface for degrading the MO solution. After adding H_2_O_2_, the degradation efficiency of Al-based glasses is improved and the Y6 ribbon has formed nano-scale crystallites embedded in the amorphous matrix and it has the largest improvement in MO solution degradation. These results indicate that forming nano-scale crystallites and adding H_2_O_2_ are effective methods to improve the degradation ability of Al-based glasses in azo dye solutions.

## 1. Introduction

In recent years, azo dyes have been widely used in domestic textile, leather, dye and other industries. However, due to its teratogenicity, carcinogenicity, chemical stability and difficult decomposition properties, it has caused serious environmental pollution problems and attracted more and more attention [1,2,3,4,5,6]. A lot of research has been done to deal with azo dye pollution, including physical adsorption of activated carbon and clays [7,8], microbial treatment [9], advanced oxidation technology [10,11,12,13] and specific alloy degradation [14,15,16,17,18,19], etc. However, these methods have obvious shortcomings, namely being short-term solutions and having low efficiency and high cost. Therefore, it is extremely urgent to explore economically and environmentally friendly advanced materials to effectively solve the pollution caused by azo dyes [20].

At present, amorphous alloys, including Fe- [21,22,23,24,25,26,27,28], Mg- [29,30,31,32], Co- [33,34,35] and Al-based [36,37,38] powders, nanoporous structures and ribbons, have been proved to have excellent degradation ability in waste water. It is generally believed that the good degradation performance of glass ribbons is due to their thermodynamic instability, easy formation of network structure and the existence of a large number of unsaturated sites on the surface. In addition, nano-scale microstructure in amorphous matrix is easy to form by annealing or reducing the cooling rate [39,40,41], which possibly has effects on the degradation and is valuable to study. The corrosion resistance of amorphous alloys is also an important factor to characterize their excellent properties [42]. The content of diffusion limitation in the kinetics involved in the catalytic degradation process has a certain influence on the degradation rate [43].

Currently, the amorphous alloys used to catalyze the degradation of MO solution or organic dyes are major in ribbon and powder shapes. Zhang et al. achieved rapid degradation of MO solution by Fenton-like reaction using Fe-based amorphous ribbons. This study found that the rate of hydroxyl radical production in Fe_78_Si_9_B_13_ amorphous ribbons was 5–10 times faster than other Fe-based catalysts [22,44,45,46,47,48]. Among all the amorphous alloy systems used for organic dye degradation, the Al-based amorphous alloys have the characteristics of low-cost materials, good reusability, high degradation efficiency, degradation under various environmental conditions, etc. Thus, Al-based amorphous alloys are a highly anticipated new method for treating organic dyes. Wang et al. studied the degradability of Al_91-x_Ni_9_Y_x_ (x = 0, 3, 6 and 9 at.%) metallic ribbons to direct blue 2B under acidic, neutral and alkaline conditions [36]. They found that Al-based alloys had a good degradation performance to direct blue 2B under different pH conditions, and the degradation efficiency decreased with the increase of element Y content.

In this paper, we studied the reactivity of Al_88_Ni_9_Y_3_ (Y3), Al_85_Ni_9_Y_6_ (Y6) and Al_82_Ni_9_Y_9_ (Y9) ribbons in degrading methyl orange (MO) azo dye solution. Under the same reaction conditions, the effect of Y content on the degradation rate of MO solution and the regulation of Y on the microstructure of Al-based alloy ribbons were studied. Under the condition of the same pH values, we explored the influence of H_2_O_2_ concentration on the degradation of MO solution. The results of this study are expected to explore the application of Al-based alloys in wastewater treatment, and study the reaction mechanism of azo dyes under different conditions, which provides a new expend direction for waste water treatment.

## 2. Experimental

### 2.1. Materials and Reagents

The alloy ingots of Al_88_Ni_9_Y_3_ (Y3, at.%), Al_85_Ni_9_Y_6_ (Y6, at.%) and Al_82_Ni_9_Y_9_ (Y9, at.%) were prepared by arc melting (MAM-1 Edmund Buhler, Berlin, Germany) of high-purity Al, Ni and Y (99.9 wt.%) metals, and the vacuum was maintained at 5 × 10^−3^ Pa and then filled with 99.999% argon. A single roller air melt spinning system (plane flow casting) was used to prepare ribbons of about 3 mm wide and 25 m thick at a speed of 44 m·s^−1^, and the ribbons were cut into 5 cm long for degradation test. Commercially available methyl orange (MO, C_14_H_14_N_3_NaO_3_S, AR grade, Tianjin Tianjin Tianxin Fine Chemical Development Center, Tianjin, China), Hydrochloric acid (HCl, AR grade, Sinopharm Chemical Reagent Co., Ltd., Shanghai, China) and Hydrogen peroxide (H_2_O_2_, AR grade, Tianjin Kemeo Chemical Reagent Co., Ltd., Tianjin, China) are used in the experiment.

### 2.2. Characterization

The microstructure of the Y3, Y6 and Y9 ribbons was characterized by X-ray diffraction and transmission electron microscopy (XRD, Bruker D8 Discover, Brooke (Beijing Technology Co., Ltd., Beijing, China; TEM, JEM-2100, Japan Electronics Co., Ltd., Beijing, China). The microstructure of the Y3, Y6 and Y9 ribbons was certified by differential scanning calorimetry (DSC, Netzsch-404, Netzsch, Bavaria, Germany). The surface morphology of the Y3, Y6 and Y9 ribbons was observed using a scanning electron microscope and X-ray energy spectrometer (SEM, EDS, JSM-7800F, Japan Electronics Co., Ltd., Beijing, China).

### 2.3. Degradation Tests

First, we prepared the MO solution (10 mg·L^−1^ MO if not noted) in a 500 mL volumetric flask with deionized water (DW) and methyl orange dye. To begin the degradation test, we poured 50 mL MO solution into a 100 mL beaker. A certain number of ribbons (0.5 g L^−1^ if not noted) and pH (pH = 1) were added to the MO solution, and the mixture was stirred at a constant rate (300 r min^−1^) during the reaction process. Then, the time interval was selected, 3 mL of the solution was extracted with a syringe and filtered with a 0.45 μm membrane; the real-time concentration of MO solution was monitored by an ultraviolet visible spectrophotometer (UV-4802, Beijing Huawei Xingye Technology Co., Ltd., Beijing, China). In the cycle test, after each degradation test, ribbons were extracted from the MO solution and cleaned with deionized water for 60 s, and then placed in the next reaction batch.

### 2.4. Electrochemical Tests

Electrochemical measuring instruments (CHI 660E, Shanghai Chenhua Instrument Co., Ltd., Shanghai, China) measured the polarization curve and impedance spectrum (EIS) in 30 mL MO solution (pH = 1, $C_{H_{2}O_{2}}$ = 1 mM and *C*_MO_ = 10 mg·L^−1^). Three electrode cells were used for the measurement; saturated calomel, platinum and ribbon were used as reference electrode, counter electrode and working electrode, respectively. When the open-circuit potential is stable, the potential scanning speed is set as 1 mV·s^−1^ to record the polarization curve. The scanning frequency was set at 100 kHz–0.01 Hz, and the amplitude was ±10 mV to record the EIS curve.

## 3. Results

### 3.1. Microstructure

Figure 1a shows the XRD patterns of the as-spun Al_88_Ni_9_Y_3_ (Y3), Al_85_Ni_9_Y_6_ (Y6) and Al_82_Ni_9_Y_9_ (Y9) ribbons. The XRD pattern of the Y3 ribbon has two crystalline peaks identified as *α*-Al and AlNi phases, and a typical diffusive scattering peak. Meanwhile, the Y3 ribbon also has *α*-Al, AlNi and amorphous cluster phases in the TEM image (Figure 1b), the results show that the Y3 ribbon has semi-amorphous structure. The XRD patterns of Y6 and Y9 ribbons show only typical diffuse scattering peak, respectively, and there is no crystal grain in the as-spun Y6 ribbon in the TEM image (Figure 1c), indicating that the Y6 and Y9 ribbons have completely amorphous structure.

Figure 1d and Table 1 show the DSC curves and thermal parameters of the as-spun Y3, Y6 and Y9 ribbons, respectively. In the DSC curves of the Y3, Y6 and Y9 ribbons, there is a crystallization onset temperature *T*_X_, three crystallization peak temperatures *T*_P1_, *T*_P2_ and *T*_P3_, a melting peak [49] temperature *T*_P4_ and a liquidus temperature *T*_L_ in the heating scan, respectively. In the DSC curves of the as-spun Y3 and Y6 ribbons, the *T*_P1_ represents the coarse crystal transition when *α*-Al phase is formed, and *T*_P2_ and *T*_P3_ represent the eutectic transition for *α*-Al/AlNi and *α*-Al/AlY phases. In the DSC curve of the as-spun Y9 ribbon, the *T*_P1_, *T*_P2_ and *T*_P3_ represent the eutectic transition for *α*-Al/AlNi, *α*-Al/AlY and *α*-Al/AlY_2_ phases, respectively. Due to the low temperature required for the coarse crystal transformation of the as-spun Y3 ribbon, a large number of *α*-Al particle clusters will be formed, thus reducing the glass formability of the as-spun Y3 ribbon. The temperature difference between coarse crystal transformation and eutectic transformation of the as-spun Y6 ribbon is small, so the *α*-Al particle clusters formed are effectively combined with the skeleton formed by eutectic transformation, and a developed network structure can be formed on the microscopic scale. Only continuous eutectic transformation occurs in the Y9 ribbon, indicating that the Y9 ribbon has good glass forming ability. In addition, the reduced glass transition temperature *T*_rX_ of the Y9 ribbon is higher than the Y3 and Y6 ribbons (Table 1), indicating that the glass formability of the Y9 ribbon is higher than the Y3 and Y6 ribbons.

### 3.2. Degradation Performance

Ultraviolet-visible absorbance spectra of filtered MO solution (*T* = 298 K, pH = 1, $C_{H_{2}O_{2}}$ = 0 mM and *C*_MO_ = 10 mg·L^−1^) after adding the Y3, Y6 and Y9 ribbons in the reaction batches for a series of time intervals (*t*_r_ = 0~45 min) are presented in Figure 2a–c, respectively. The spectrum of the MO solution at about 508 nm has a main absorption peak, which represents the chromophore (-N=N-) of the MO dye. Take the peak value at 508 nm to obtain the normalized concentration of the MO solution, as shown in Figure 2d. In the reaction with the as-spun Y3, Y6 and Y9 ribbons, the concentration of the solution remained almost unchanged in the first 5 min and then decreased rapidly. With the increase of reactive time *t*_r_. The absorption peak gradually attenuates, indicating that the MO solution is gradually degraded. The degradation kinetics after 5 min is generally described by quasi-first-order equation [50,51]:*C*_t_ = *C*_0_ exp(−*kt*_r_)(1)
where *k* is the reaction rate constant (min^−1^), *C*_0_ is the initial concentration of the MO solution (mg·L^−1^) and *C*_t_ is the instant concentration of the MO solution (mg·L^−1^) at *t*_r_. Then, the degradation reaction rate constant can be derived as follows:(2)$$k=\;\ln\;\left(\frac{C_{0}}{C_{t}}\right)/t_{r}$$

According to the ln (*C*_0_*/C*_t_)-*t*_r_ curves is shown in the lower left illustration of Figure 2d, the reaction rate constant of the Y3 ribbon is 0.034 min^−1^, which is larger than the Y6 (0.023 min^−1^) and Y9 (0.019 min^−1^) ribbons. Here, the fit goodness values *R*^2^ of the Y3, Y6 and Y9 ribbons are 0.97, 0.98 and 0.98, respectively. Thus, the Y3 ribbon exhibits a better catalytic degradation efficiency for the MO solution than the Y6 and Y9 ribbons.

### 3.3. Effect of H_2_O_2_ Concentration

The influence of the H_2_O_2_ concentration as well as XRD patterns of the Y3, Y6 and Y9 ribbons on the reaction process of the MO solution was studied, as shown in Figure 3. In the MO solution, different concentrations of 0, 0.5 and 1 mM of H_2_O_2_ were added, and the other reaction conditions were constant: *T* = 298 K, pH = 1, ribbon dosage = 0.5 g·L^−1^ and *C*_MO_ = 10 mg·L^−1^. When the H_2_O_2_ = 0, 0.5 and 1 mM, the difference value of the degradation efficiency *η* (*η* = (1 − *C*_t_/*C*_0_ × 100%, *t* = 45 min) of the Y3 ribbon on MO solution increased gradually (Figure 3a) and the as-spun Y6 ribbon increased relatively steadily (Figure 3b), while the as-spun Y9 ribbon decreased gradually (Figure 3c). It is proved that H_2_O_2_ can effectively increase the degradation of MO solution. When the $C_{H_{2}O_{2}}$ is 1 mM, the hydrogen ions and H_2_O_2_ in MO solution can generate nascent hydrogen [H] and •OH groups with aluminum and other metallic on the surface, thus accelerating the degradation of MO solution. As the $C_{H_{2}O_{2}}$ = 0.5 mM, the content of H_2_O_2_ in MO solution is lower, and the number of generated •OH groups will decrease, thus reducing the *η* of MO solution. As the $C_{H_{2}O_{2}}$ = 0 mM, the degradation of MO solution can only be achieved by the generated [H], so the degradation efficiency is lower than that when $C_{H_{2}O_{2}}$ is 0.5 and 1 mM.

Figure 3d–f show the XRD patterns of different states of the Y3, Y6 and Y9 ribbons, respectively. Here, the as-spun ribbons, reacted (*t*_r_ = 45 min) in MO solution with $C_{H_{2}O_{2}}$ = 0, 0.5 and 1 mM, are labeled as Y3/Y6/Y9_as_, Y3/Y6/Y9_0mM_, Y3/Y6/Y9_0.5mM_ and Y3/Y6/Y9_1mM_, respectively. As shown in Figure 3d, the crystalline peaks of Y3_0mM_, Y3_0.5mM_ and Y3_1mM_, identified as *α*-Al and AlNi phases, are lower than the Y3_as_ ribbon; moreover, the crystalline peak of the *α*-Al phase gradually decreases with the increase of $C_{H_{2}O_{2}}$ = 0, 0.5 and 1 mM, and the AlNi phase is almost completely consumed. Thus, it can be seen that when reacted with the MO solution for 45 min, the fraction of crystalline *α*-Al and AlNi phases on the Y3 ribbon surface reduced. As shown in Figure 3e, the Y6_as_, Y6_0mM_, Y6_0.5mM_ and Y6_1mM_ all have a typical diffusive scattering peak. However, we observed that with the increase of $C_{H_{2}O_{2}}$, the microstructure of Y6_0mM_, Y6_0.5mM_ and Y6_1mM_ ribbons changed slightly, and a certain number of crystalline peaks appeared, which gradually strengthened with the increase of $C_{H_{2}O_{2}}$. As shown in Figure 3f, with the increase of $C_{H_{2}O_{2}}$, the Y9_0mM_, Y9_0.5mM_ and Y9_1mM_ ribbons also show some crystalline peaks, but the crystallization strength is lower than that of Y6_0mM_, Y6_0.5mM_ and Y6_1mM_ ribbons, respectively, which indicates that the Y6_0mM_/Y6_0.5mM_/Y6_1mM_ and Y9_0mM_/Y9_0.5mM_/Y9_1mM_ ribbons have crystallized on the surface of the ribbon during the catalytic degradation of the MO solution, and the stability of the amorphous Y9_as_ ribbon is higher than that of the Y6_as_ ribbon.

### 3.4. Surface Morphology

To understand the MO solution (*T* = 298 K, pH = 1, $C_{H_{2}O_{2}}$ = 1 mM and *C*_MO_ = 10 mg·L^−1^) degradation process with the as-spun Y3, Y6 and Y9 ribbons, it is of great significance to study the structural evolution of ribbon surface after reaction. SEM images of as-spun and reacted surfaces of Y3, Y6 and Y9 ribbons are shown in Figure 4, and EDS results are shown in Table 2. Typical smooth surfaces are observed on the Y3, Y6 and Y9 ribbons, as shown in Figure 4a–c, respectively. The reacted Y3 ribbon surface appears to have large nanometer channels (lower left inset in Figure 4d), and the reacted surface of the Y6 ribbon has a developed network structure (lower left inset in Figure 4e). There is a fuzzy network structure and a corrosion pit on the surface of the reacted Y9 ribbon (lower left inset in Figure 4f). After 45 min of reaction, *c*_Al_ in the Y3, Y6 and Y9 ribbons reduced, and the *c*_Ni_ and *c*_Y_ were unchanged. At the same time, the decrease amplitude of Al on the surface of Y3 ribbon is higher than that of Y6 and Y9 ribbons, which indicates that the high reaction rate of the Y3 ribbon is due to the presence of a large amount of Al involved in the degradation reaction. After degradation, *c*_O_ on the surface of Y3, Y6 and Y9 ribbons increased, which indicated that the degradation process involved the oxidation of ribbons.

### 3.5. Electrochemical Analysis

The electrochemical properties of the Y3, Y6 and Y9 ribbons in MO solution are studied, and are shown in Figure 5. The corrosion potential (*E*_corr_) of the as-spun Y3 ribbon is −0.32 V, which is higher than of the as-spun Y6 (−0.36 V) and Y9 (−0.38 V) ribbons (Figure 5a); in addition, the corrosion current densities (*i*_corr_) of the as-spun Y3 ribbon is 2.04 × 10^−6^ A cm^−2^, which is lower than of the Y6 (9.98 × 10^−6^ A cm^−2^) and Y9 (1.67 × 10^−5^ A cm^−2^) ribbons. The above polarization curve data show that the corrosion resistance of the as-spun Y3 ribbon is higher than that of the Y6 and Y9 ribbons. The above polarization curve data show that compared with the Y6 and Y9 ribbons, the Y3 ribbons have higher corrosion resistance. Figure 5b shows that the Nyquist semicircle diameter of the Y3 ribbon is larger than the Y6 and Y9 ribbons. An equivalent circuit composed of R(C(R(C(R(CR))))) was used to fit EIS data. The chi square (χ^2^) of the Y3, Y6 and Y9 ribbons are 1.08 × 10^−3^, 1.09 × 10^−3^ and 5.61 × 10^−4^, respectively. Furthermore, the phasing element (CPE) *Q* is defined as [42]:(3)$$Q={\left(jw\right)}^{-n}\;/\;Y_{0}$$

Table 3 shows the fitting results. It can be seen that the total resistance (*R*_total_) of the Y3 ribbon is higher than the Y6 and Y9 ribbons. Therefore, the results of EIS are consistent with those of polarization curve (Figure 5a). Interestingly, the higher corrosion resistance of the Y3 ribbon has no harmful effect on MO solution degradation and is possibly beneficial for the durability in the degradation process, which is valuable for further study.

### 3.6. Reusability of Y3 in Degradation MO Solution

In order to further study the reusability of the ribbons, we selected the Y3 ribbon to repeatedly degrade the MO solution under the condition of *T* = 298 K, pH = 1, $C_{H_{2}O_{2}}$ = 1 mM and *C*_MO_ = 10 mg·L^−1^, as shown in Figure 6. The degradation efficiency *η* of the Y3 ribbon on the MO solution decreased slowly in the 1st to the 3rd test, and decreased greatly in the 4th test. Compared with the 4th test, the *η* of the 5th test remained basically unchanged, and still has relatively high degradation efficiency. The normalized concentration *C*_t_/*C*_0_ of the MO solution reacting of the Y3 ribbon remained almost unchanged during the first 5 min; this may be because in the 1st test, it takes some time to consume the oxide on the surface of the ribbon, and for the fresh metal element surface to react, so the degradation reaction can be achieved quickly in the 2nd to 5th tests. In addition, we found that the degradation rate *k* of the 2nd test was higher than that of the 1st test, which was also related to the elimination of oxide film.

## 4. Discussion

After the microstructure analysis of the as-spun Y3, Y6 and Y9 ribbons, it was found that both the nano-crystalline phase and amorphous cluster phase exist in the as-spun Y3 ribbon, while only the amorphous cluster phase exists in the as-spun Y6 and Y9 ribbons (Figure 1). Since there are many nanometer phase grain boundaries in the nanometer phase of the as-spun Y3 ribbon, a large number of nanometer channels will be formed when these grain boundaries contact with the amorphous cluster phase [52,53,54]. These channels can effectively transport substances, and transfer the elements inside the ribbon to the surface to participate in the reaction. Although the as-spun Y6 and Y9 ribbons are fully amorphous ribbons, according to the DSC curves (Figure 1d), the as-spun Y6 ribbon easily forms a network structure on a microscopic scale, while the as-spun Y9 ribbon has a good amorphous forming ability and stability.

With increasing *c*_Y_, the as-spun Y3, Y6 and Y9 ribbons have a gradually decreased degradation rate in the MO solution (Figure 2d), indicating that the increase of the Y element will inhibit the generation of [H] and •OH groups [55,56,57,58]. According to the results of the XRD pattern (Figure 1a) and TEM image (Figure 1b), the as-spun Y3 ribbon confirmed the formation of *α*-Al and AlNi phases. The DSC curve showed that at 435 K (Figure 1d and Table 1), the coarse crystal transformation occurred to form *α*-Al particle cluster phase. After the as-spun Y3 ribbon reacted with MO solution (*T* = 298 K, pH = 1, $C_{H_{2}O_{2}}$ = 1 mM and *C*_MO_ = 10 mg·L^−1^) for 45 min, a large number of nanometer channels appeared on the surface (Figure 4d), these channels were caused by the Al element in the *α*-Al particle cluster phase escaping and participating in the degradation of the MO solution. The as-spun Y6 and Y9 ribbons are fully amorphous ribbons. According to the DSC curve, we find that the Y6 ribbon still have coarse crystal phase *α*-Al particle clusters at 520 K. During the degradation process, some particles gradually appear on the surface of the ribbons, and these particles combine with each other to form a developed network structure [59], which can be confirmed in Figure 4e. In the DSC curve, the as-spun Y9 ribbon only had eutectic transformation, so it has a high amorphous forming ability, and the formed microscopic network structure tends to be stable. In the process of reaction with the MO solution, only fuzzy network structure is formed on the ribbon surface (Figure 4f).

Under different $C_{H_{2}O_{2}}$, the as-spun Y3, Y6 and Y9 ribbons have different degradation efficiency *η* in the MO solution (Figure 3a–c). As $C_{H_{2}O_{2}}$ is 0, 0.5 and 1 mM, the degradation efficiency of the Y3 ribbon is higher than that of the Y6 and Y9 ribbons, which may be related to the fact that the Y3 ribbon is semi-amorphous. The crystalline phases *α*-Al and AlNi in the Y3 ribbon can release the Al element to participate in the degradation reaction when degrading the MO solution. After the as-spun Y3 ribbon reacted in the MO solution with $C_{H_{2}O_{2}}$ is 0, 0.5 and 1 mM, the crystalline phase strength of the ribbons gradually decreased with the increase of $C_{H_{2}O_{2}}$ (Figure 3d), and correspondingly, the degradation efficiency of the Y3 ribbon in the MO solution gradually increased (Figure 3a), which confirmed that the Al and AlNi crystalline phase can accelerate the degradation of the MO solution. With regard to the as-spun Y6 and Y9 fully amorphous ribbons, the Y6 ribbon has higher degradation efficiency than the Y9 ribbon (Figure 3b,c), which is because the developed network structure formed by the Y6 ribbon during the degradation of the MO solution can provide more active sites (Figure 4e), therefore accelerating the catalytic degradation of the MO solution. In addition, the XRD patterns of Figure 3e,f show that with the increase of $C_{H_{2}O_{2}}$, the reacted Y6 and Y9 ribbons will show different crystalline peaks, and the crystalline peak intensity of the Y6 ribbon is higher than the Y9 ribbon. Thus, we think that the developed network structure and crystalline peak on the surface of the Y6 ribbon are the main factors for the degradation efficiency higher than the Y9 ribbon.

Based on the above analysis, the surface micro-morphology, surface element information and microstructure of the Y3 and Y6 ribbons during the reaction process of the MO solution were analyzed, we have drawn the schematic diagram as shown in Figure 7. When the $C_{H_{2}O_{2}}$ = 1 mM in the MO solution, the ribbons on the degradation of the MO solution can be divided into two ways. First, the metallic aluminum ionizes to generate electrons, which then combine with hydrogen ions to generate [H] with reducibility [55]. Second, the metallic aluminum reacts with H_2_O_2_ in solution to produce •OH group with oxidizability [58], and we find that the [H] plays a dominant role in the degradation of MO solution. During the degradation of the MO solution, the Y3 and Y6 ribbons gradually formed developed nanometer channels and network structures on their surfaces respectively, the reaction resistance (*R*_a_) of Y3 ribbon is greater than that of Y6 ribbon. The formation of nanometer channels in the Y3 ribbon is due to the existence of the nano-crystalline phase in the ribbon, and the appearance of a network structure on the surface of the Y6 ribbon is related to the skeleton in the amorphous matrix. The nanometer channels can continuously transport Al elements to the surface to participate in the reaction, and the specific surface area of the reaction is increased by the surface network structure, so they can accelerate the catalytic degradation of the MO solution.

## 5. Conclusions

In this work, we have melt-spun Al_88_Ni_9_Y_3_ (Y3), Al_85_Ni_9_Y_6_ (Y6) and Al_82_Ni_9_Y_9_ (Y9) glassy ribbons. Furthermore, we studied their microstructure, such as nano-scale crystallites, and the MO solution degradation process by adding H_2_O_2_. We have found:(1)With increasing *c*_Y_, the as-spun Y3, Y6 and Y9 ribbons have an increasing GFA (glass formability) and gradually decreased the degradation rate of the MO solution. These results indicate that the nano-scale crystallites in the Y3 ribbon can form the nanometer channels to transport elements to the surface for degrading the MO solution.(2)After adding H_2_O_2_, the degradation efficiency of the Al-based glasses is improved. The Y6 ribbon has the largest improvement, which is ascribed to the formation of nano-scale crystallites in the degradation and then developed network structure on the sample surface.

## Figures and Tables

**Figure 1 materials-14-00039-f001:**
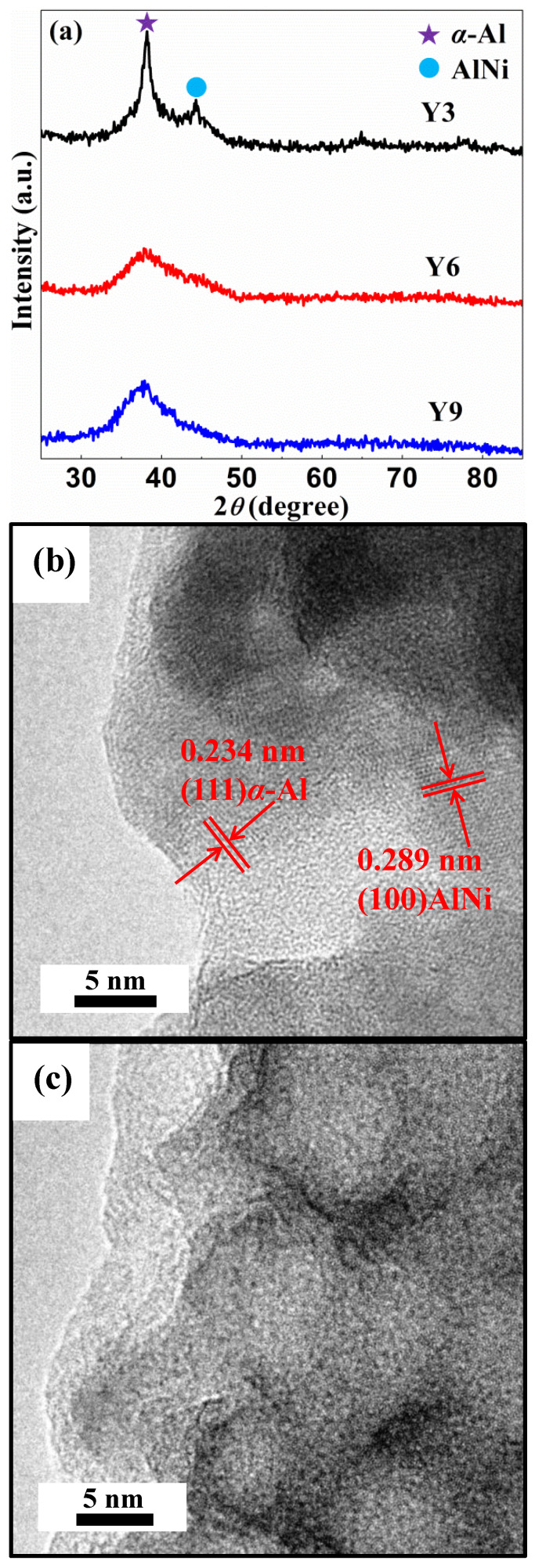
(**a**) The XRD patterns of the as-spun Al_88_Ni_9_Y_3_ (Y3), Al_85_Ni_9_Y_6_ (Y6) and Al_82_Ni_9_Y_9_ (Y9) ribbons, the TEM images of as-spun (**b**) Y3 and (**c**) Y6 ribbons and (**d**) the DSC curves of the as-spun Y3, Y6 and Y9 ribbons.

**Figure 2 materials-14-00039-f002:**
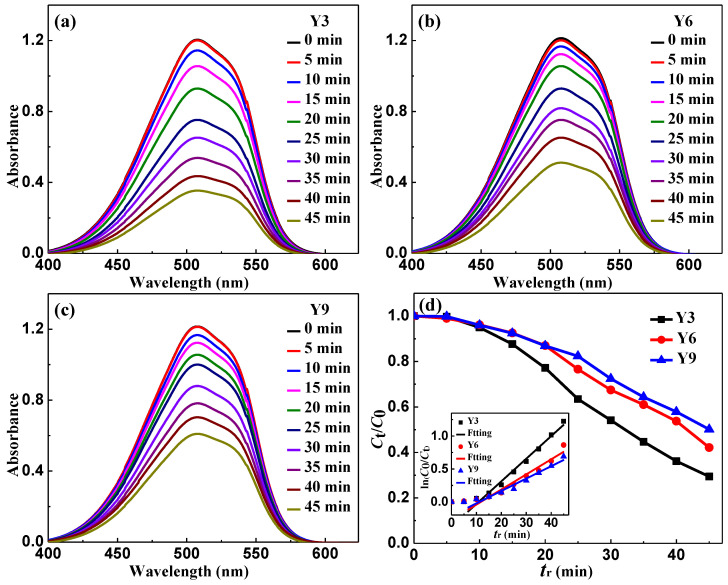
UV-Vis absorbance spectra of the as-spun (**a**) Al_88_Ni_9_Y_3_ (Y3), (**b**) Al_85_Ni_9_Y_6_ (Y6) and (**c**) Al_82_Ni_9_Y_9_ (Y9) ribbons after they reacted with the MO solution (*T* = 298 K, pH = 1, $C_{H_{2}O_{2}}$ = 0 mM and *C*_MO_ = 10 mg·L^−1^) and (**d**) normalized concentration change of the MO solution during the degradation process. The lower left inset in (**d**): the ln(*C*_0_/*C*_t_)-*t*_rea_ curves for the as-spun Y3, Y6 and Y9 ribbons. The symbols show the experimental data while the solid lines are the fitting results.

**Figure 3 materials-14-00039-f003:**
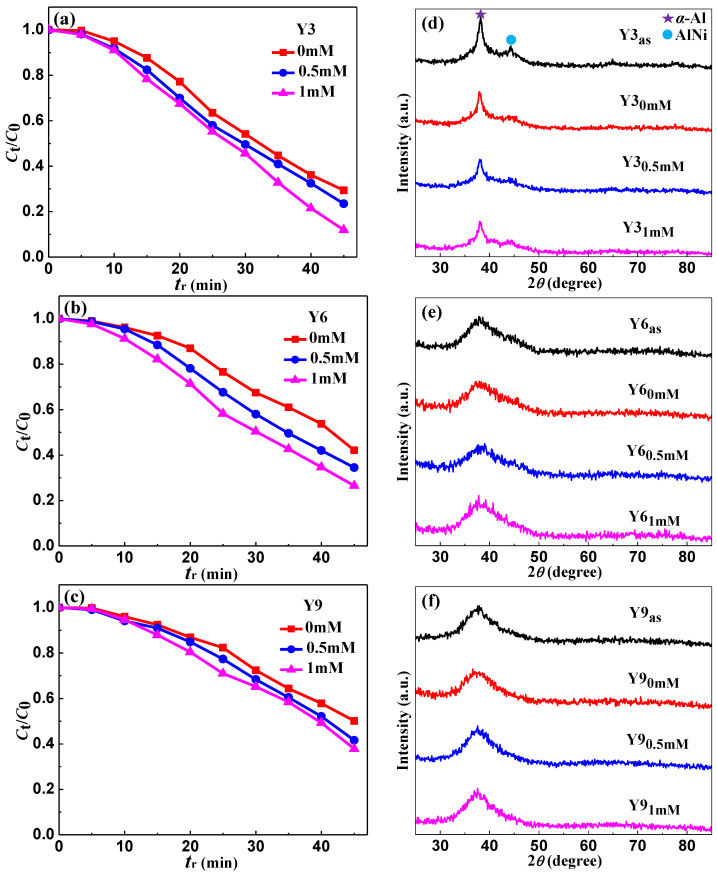
The normalized concentration *C*_t_/*C*_0_ change of MO solution during the degradation process of the as-spun (**a**) Al_88_Ni_9_Y_3_ (Y3), (**b**) Al_85_Ni_9_Y_6_ (Y6) and (**c**) Al_82_Ni_9_Y_9_ (Y9) ribbons at different H_2_O_2_ concentration and the XRD patterns of the (**d**) Y3, (**e**) Y6 and (**f**) Y9 ribbons in different states.

**Figure 4 materials-14-00039-f004:**
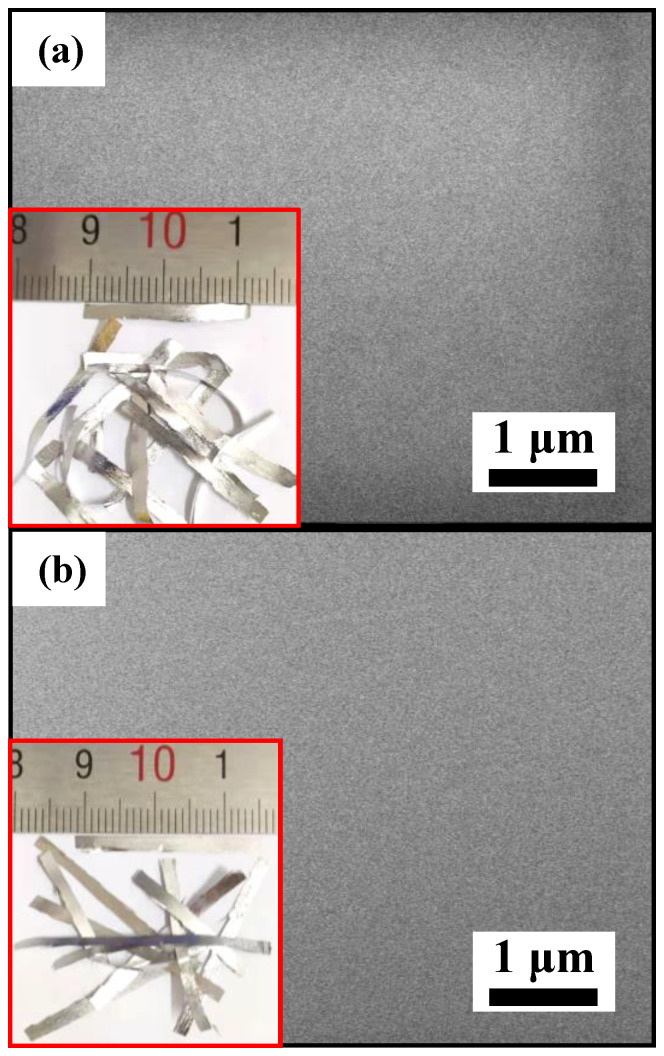
SEM micrographs of the as-spun (**a**) Al_88_Ni_9_Y_3_ (Y3), (**b**) Al_85_Ni_9_Y_6_ (Y6) and (**c**) Al_82_Ni_9_Y_9_ (Y9) ribbons, and the reacted (**d**) Y3, (**e**) Y6 and (**f**) Y9 ribbons. The lower left insets in (**a**–**c**): the as-spun Y3, Y6 and Y9 ribbons physical photos. The lower left insets in (**d**–**f**): the high-magnification images of reacted Y3, Y6 and Y9 ribbons.

**Figure 5 materials-14-00039-f005:**
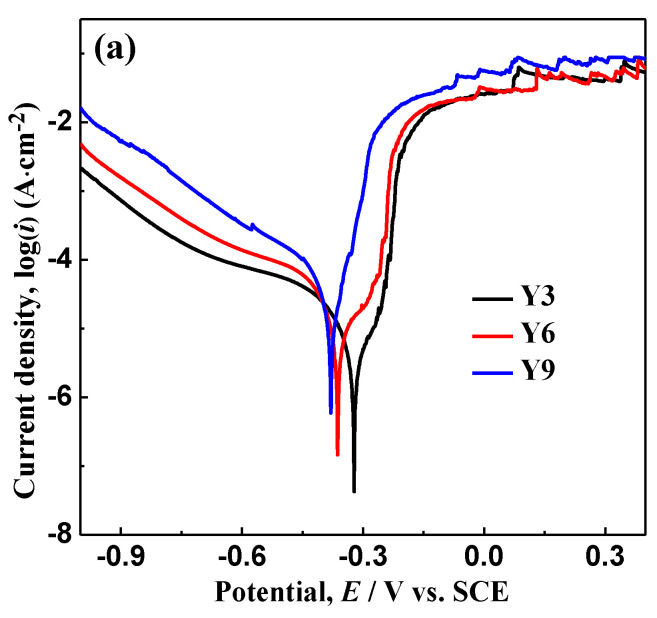
(**a**) Polarization curves and (**b**) Nyquist curves of the as-spun Al_88_Ni_9_Y_3_ (Y3), Al_85_Ni_9_Y_6_ (Y6) and Al_82_Ni_9_Y_9_ (Y9) ribbons in the MO solution (*T* = 298 K, pH = 1, $C_{H_{2}O_{2}}$ = 1 mM and *C*_MO_ = 10 mg·L^−1^). The upper insets in (**b**): the general fitted circuit. Symbols represent experimental data and solid lines represent fitting results.

**Figure 6 materials-14-00039-f006:**
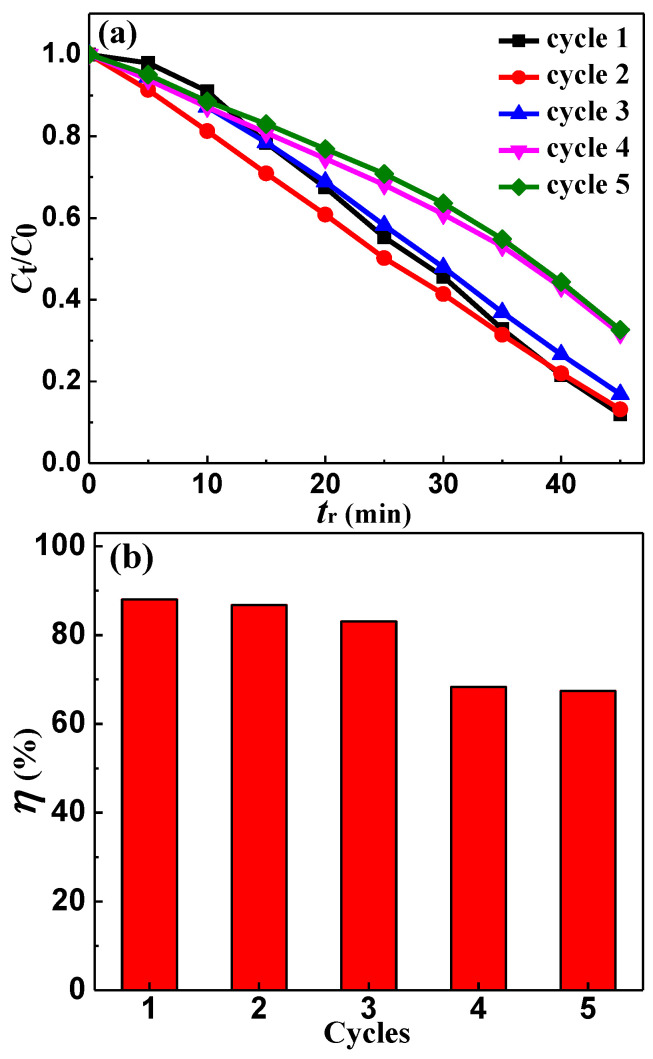
(**a**) The normalized *C*_t_/*C*_0_ change of the MO solution during the degradation process of the Al_88_Ni_9_Y_3_ (Y3) ribbon from the 1st to the 5th cycle. (**b**) The degradation efficiency (*η* = (1 − *C*_t_/*C*_0_ × 100%, *t*_r_ = 45 min) of the degradation process vs. the reaction cycles for the Y3 ribbon.

**Figure 7 materials-14-00039-f007:**
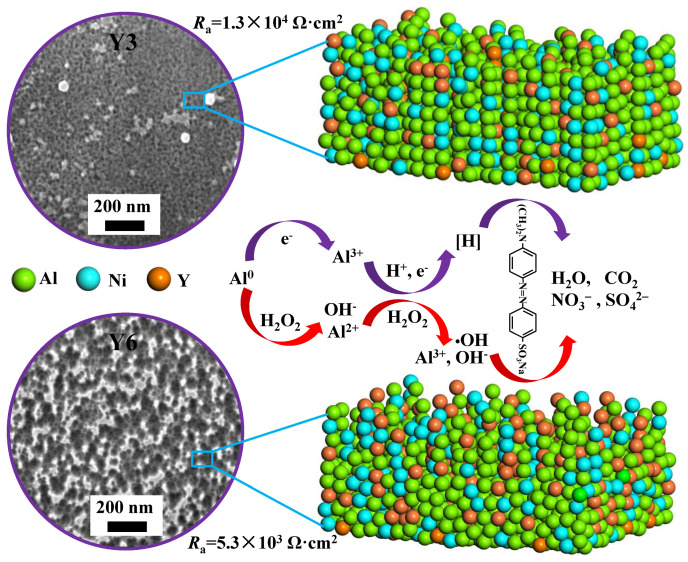
Schematic illustration of the degradation mechanism of the MO dyes using the Al_88_Ni_9_Y_3_ (Y3) and Al_85_Ni_9_Y_6_ (Y6) ribbons.

**Table 1 materials-14-00039-t001:** DSC analysis of the Al_88_Ni_9_Y_3_ (Y3), Al_85_Ni_9_Y_6_ (Y6) and Al_82_Ni_9_Y_9_ (Y9) ribbon thermal parameters.

Alloy	Temperature (K)						
	*T* _X_	*T* _P1_	*T* _P2_	*T* _P3_	*T* _P4_	*T* _L_	*T*_rX_ (*T*_X_/*T*_L_)
Y3	407	435	613	657	919	959	0.42
Y6	504	520	602	636	918	958	0.53
Y9	571	577	632	759	917	956	0.60

**Table 2 materials-14-00039-t002:** EDS analysis of the Al_88_Ni_9_Y_3_ (Y3), Al_85_Ni_9_Y_6_ (Y6) and Al_82_Ni_9_Y_9_ (Y9) ribbons before and after they reacted (at.%).

Alloy	Before				After			
	*c* _Al_	*c* _Ni_	*c* _Y_	*c* _O_	*c* _Al_	*c* _Ni_	*c* _Y_	*c* _O_
Y3	87.6	8.5	3.0	0.9	85.9	8.6	3.1	2.4
Y6	83.4	10.1	5.3	1.2	82.7	9.9	5.4	2.0
Y9	80.7	8.2	8.2	1.2	80.0	9.7	9.2	1.2

**Table 3 materials-14-00039-t003:** Parameters from EIS measurements: *R*_s_, solution resistance; *C*_dl_, resistance of electric double layer; *R*_t_, resistance of transfer charge; *C*_f_ and *R*_f_, resistance of passivation film; *C*_a_ and *R*_a_, resistance of electrochemical reaction; *R*_total_, total resistance.

Alloy	*R*_s_ (Ω·cm^2^)	*C*_dl_ (Ω^−1^·cm^−2^)	*R*_t_ (Ω·cm^2^)	*C*_f_ (Ω^−1^·cm^−2^)	*R*_f_ (Ω·cm^2^)	*C*_a_ (Ω^−1^·cm^−2^)	*R*_a_ (Ω·cm^2^)	*R*_total_ (Ω·cm^2^)
Y3	2.9	1.1 × 10^−6^	305.5	1.2 × 10^−6^	7.1 × 10^−3^	3.4 × 10^−10^	1.3 × 10^4^	13,308
Y6	2.3	6.8 × 10^−6^	56.5	4.3 × 10^−6^	1038.3	3.5 × 10^−3^	5.3 × 10^3^	6397
Y9	2.4	1.9 × 10^−6^	107.0	1.2 × 10^−6^	652.3	1.3 × 10^−6^	4.7 × 10^3^	5462

## Data Availability

Data available in a publicly accessible repository that does not issue DOIs.

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
