# Peer review of "Role of Nanocrystallites of Al-Based Glasses and H2O2 in Degradation Azo Dyes"

_materials, 2020, doi:10.3390/ma14010039_

Round 1

Reviewer 1 Report

The paper presents the factors affecting the degradation of methyl orange using Al-based metallic glasses. The data presented are extensive and the explanations are complete. Hence, I recommend for the acceptance of the manuscript pending moderate English language revision. Also, all abbreviations must also be defined when first used in the text. In section 2.3, what does if not noted means?

Author Response

The paper presents the factors affecting the degradation of methyl orange using Al-based metallic glasses. The data presented are extensive and the explanations are complete. Hence, I recommend for the acceptance of the manuscript pending moderate English language revision. Also, all abbreviations must also be defined when first used in the text. In section 2.3, what does if not noted means?

Response:We have carefully revised the English grammar and structure of this manuscript, and all abbreviations were defined when they first appeared in the manuscript, which is marked with red underlines. The "if not noted" means that if there is no special explanation elsewhere in the manuscript, it will be considered as the corresponding content in section 2.3.

Reviewer 2 Report

Dear authors

The paper presented is of very high standrads, and open many doors to follow for readers and researchers. The amount of experimental results is extense enough, and I want to profit the ocasion to congratulate you for a so interesting work. Nevertheless, I have some comments to do:

  • In general aspects, Introduction must to be improved with the addition of the aim of the electrochemical experiments done. The interest about to detemrine the polarization, should be stated, clearly. Moreover, kinetics should include the diffusion-limited aspects that are the controlling phenomena of the reaction. When dyes are contacted with Y's species, without hydrogen peroxyde, would be absorbed by Y's and that, probably is the reason of the decreasing of concentration of dye into the bath, and not reaction. At least that aspect of the experimental results should be clarified.
  • Some more pictures related with reactions and phenomena involved should be add in order to state the importance of polarization in the steps of reactions. (the swume is in Fig. 6, at the end of the explanations)
  • Please, consider to change some pictures that doesn't give the information described from the experimental results (Fig 4, a, b, c)
  • In catalytic reactions like these, the size of the particle is absolutely necessary to be used, because the kinetic constant calculated, will include the geometrical factors of them.Please, consider to comment and to revise some of the discussions from this point of view

In more specific aspects:

-In introduction, please consider to revise the first paragraph, I have detected some grammatical mistakes

Page 3: First paragraph. It keeps not clear wich is the aim of the experiment described when dealing with the polarization curves in via radical reactions

Please, consdier to make a picture that helps to imagine the geometry of "ribbons"

Page 4, eq 1: Include some diffusional aspects in the asumption of a first order reaction. There are some bibliography related with kinetics on heterogeneous systems that will help

Fig 3 (b) there is a different behaviour in this case with Y6 behaviour with the increase of peroxyde concentration, this is a behaviour that would be interesting to explain, because it is a typical diffusion controlled reactions system

The attribution to H nascient and radical hydroxyl is very interesting and, in my opinion should deserve more attention 

Many thanks

Author Response

Point 1: In general aspects, Introduction must to be improved with the addition of the aim of the electrochemical experiments done. The interest about to detemrine the polarization, should be stated, clearly. Moreover, kinetics should include the diffusion-limited aspects that are the controlling phenomena of the reaction.

Response 1: The purpose of the electrochemical test on the ribbon is to study the corrosion resistance of Y3, Y6 and Y9 ribbons, and the presence of diffusion -limited in the reaction process would also affect the degradation rate constant k. In the introduction, we add the content of electrochemical experiments and the diffusion limitation knowledge related to kinetics, and added relevant references [42, 43], which is marked with red underlines.

Ref [42]: Jia, C.G.; Pang, J.; Pan, S.P.; Zhang, Y.J.; Kim, K.B.; Qin, J.Y.; Wang, W.M., Tailoring the corrosion behavior of Fe-based metallic glasses through inducing Nb-triggered netlike structure. Corros. Sci. 2019, 147, 94-107.

[43]: Hakan, B.; Benjamin, J.M.; Timothy, R.G.; Frank, J.L.; Dietrich, J.P., A diffusion limited sorption kinetics model with polydispersed particles of distinct sizes and shapes. Adv. Water Resour. 2002, 25, 755-772.

Point 2: When dyes are contacted with Y's species, without hydrogen peroxyde, would be absorbed by Y's and that, probably is the reason of the decreasing of concentration of dye into the bath, and not reaction. At least that aspect of the experimental results should be clarified.

Response 2: We believe that the adsorption capacity of the ribbon to the dye molecules is very small and the adsorbed dye molecules can be quickly degraded by the catalyst. The Y3, Y6 and Y9 ribbons in the MO solution without hydrogen peroxide can degrade the dye by generating nascent hydrogen [H], and the addition of hydrogen peroxide will produce •OH to accelerate the dye degradation. We have clearly explained the degradation mechanism in the discussion section of the manuscript.

Point 3: Some more pictures related with reactions and phenomena involved should be add in order to state the importance of polarization in the steps of reactions. (the swume is in Fig. 6, at the end of the explanations)

Response 3: Through the polarization curve test, we found that the increase of Y element content reduced the corrosion resistance of the ribbon, and the reaction resistance (Ra) with Y3, Y6 and Y9 ribbons involved in the reaction were obtained in the impedance spectra (EIS) test. In Fig. 6, the surface morphology of Y3 and Y6 ribbons after reaction is compared. Compared with Y3 ribbon, the surface of Y6 ribbon is more concave and convex, indicating that Y6 ribbon is more prone to corrosion during the reaction, and we add the reaction resistance to Fig. 6 to make Fig. 6 more meaningful, which is well corresponding to the polarization curve.

Point 4: Please, consider to change some pictures that doesn't give the information described from the experimental results (Fig 4, a, b, c)

Response 4: Fig. 4a, b and c are given in order to make a better comparison with the ribbon after the reaction and facilitate the reader's understanding. And in Fig. 4a, b and c, we inserted the physical photos with Y3, Y6 and Y9 ribbons.

Point 5: In catalytic reactions like these, the size of the particle is absolutely necessary to be used, because the kinetic constant calculated, will include the geometrical factors of them. Please, consider to comment and to revise some of the discussions from this point of view.

Response 5: In the catalytic reaction, particles with different sizes are formed on the surface of different ribbons, which together form a network structure on the surface of the ribbons, which can provide more sites for the reaction and accelerate the degradation rate. When we calculate the kinetic constant k, we get the apparent rate constant, and the available data do not reflect the influence of the geometrical factors. The influence of particle size on the degradation rate constant is added in the discussion section and added relevant references [59], which is marked with red underlines.

Ref [59]: Giona, M.; First-order reaction-diffusion kinetics in complex fractal media. Chem. Eng. Sci. 1992, 47, 1503-1515.

Point 6: In introduction, please consider to revise the first paragraph, I have detected some grammatical mistakes

Response 6: We have carefully revised the English grammar and structure of the first paragraph, which is marked with red underlines.

Point 7: Page 3: First paragraph. It keeps not clear wich is the aim of the experiment described when dealing with the polarization curves in via radical reactions

Response 7: In the manuscript, we concluded that when hydrogen peroxide was added, the degradation rate of MO solution was improved by the ribbon. We wanted to test the polarization curve to find out the influence of Y element content on the corrosion resistance of the ribbon, and obtain their specific corrosion resistance through impedance spectra (EIS) test, so as to verify the effectiveness of the polarization curve.

Point 8: Please, consdier to make a picture that helps to imagine the geometry of "ribbons"

Response 8: In the manuscript, we have supplemented the ribbon photos of Y3, Y6 and Y9 in Fig. 4a, b and c.

Figure 4. SEM micrographs of the as-spun (a) Al88Ni9Y3 (Y3), (b) Al85Ni9Y6 (Y6) and (c) Al82Ni9Y9 (Y9) ribbons, the lower left insets in (a), (b) and (c): the as-spun Y3, Y6 and Y9 ribbons physical photos.

Point 9: Page 4, eq 1: Include some diffusional aspects in the asumption of a first order reaction. There are some bibliography related with kinetics on heterogeneous systems that will help

Response 9: The Y3, Y6 and Y9 ribbons on the degradation rate of MO solution, we use first-order equation calculation, the different ribbon due to different degradation rate, with the ribbon with fractal dimension in the process of reaction so as to make the surface reticular structure, different reticular structure makes the ribbon for the spread of the degradation of MO solution with different limit, so have different degradation rate. The ribbon catalytic degradation of MO solution is a heterogeneous reaction, and the reaction rate is related to the contact area of the phase interface. We added references [51] to heterogeneous systems to the manuscript.

Ref [51]: Geng, N.N.; Chen, W.; Xu, H.; Ding, M.M.; Lin, T.; Wu, Q.S.; Zhang, L., Insights into the novel application of Fe-MOFs in ultrasound-assisted heterogeneous Fenton system: Efficiency, kinetics and mechanism. Ultrason. Sonochem. 2021, 72, 105411.

Point 10: Fig 3 (b) there is a different behaviour in this case with Y6 behaviour with the increase of peroxyde concentration, this is a behaviour that would be interesting to explain, because it is a typical diffusion controlled reactions system

Response 10: When Y6 ribbon degrade MO solution, different concentrations of hydrogen peroxide have great differences in the degradation rate. High concentration of hydrogen peroxide is more likely to diffuse to the active site involved in the reaction, and the formation of degradation groups accelerates the degradation rate. In addition, different concentrations of hydrogen peroxide formed different network structures on the surface of the ribbon through diffusion control reaction, and the increase of the reaction site made the degradation rate of MO solution significantly different.

Reviewer 3 Report

This manuscript reports the role of nanocrystallites of Al-based glasses and H2O2 in the degradation of azo dyes. The study is of reasonable quality and the article is interesting. The work is clear and well written. It may be of interest to some readers and may be publishable.

Author Response

This manuscript reports the role of nanocrystallites of Al-based glasses and H2O2 in the degradation of azo dyes. The study is of reasonable quality and the article is interesting. The work is clear and well written. It may be of interest to some readers and may be publishable.

Response: Thank the reviewers for their comments on our work, and I would like to extend my sincere greetings to you.

Reviewer 4 Report

The differences of Al-based amorphous alloys (Y3, Y6, and Y9) were used for the degradation of methyl orange (MO) azo dye solution with/without adding H2O2. The highest degradation efficiency of MO was achieved by Y3 which is a semi-amorphous structure, compared with Y6 and Y9 which are fully amorphous structures. The increasing of H2O2 dosage improves the degradation activity of MO. Also, the surface morphology and phase structure were well characterized. Thus, I think the manuscript can be accepted for publishing after revision by the following comments:

  1. The authors should explain in more detail the currently used catalyst for MO or organic degradations.
  2. Please discuss the relationship between the formation of the crystal phase of Y3 compared with Y6 and Y9 and different Al and Y contents.
  3. The authors should give references to the melting peak in DRS curves.
  4. The authors should explain why the concentration of the MO solution remained almost unchanged in the first 5 min of the reaction.
  5. The authors mention that the [H] plays a dominant role in the degradation of MO solution. Thus, The authors should provide some evidence to prove this statement. However, it is not consistent with the explanation of the effect of H2O2 dosage which describes the importance of hydroxyl radical from H2O2.
  6. The authors should test the reusability for the MO degradation over the catalyst.

Author Response

Point 1: The authors should explain in more detail the currently used catalyst for MO or organic degradations.

Response 1: In the introduction, we added catalysts for the current degradation of MO and organics, Zhang et al. achieved rapid degradation of MO solution by Fenton-like reaction using Fe-based amorphous ribbons. This study found that the rate of hydroxyl radical production in Fe78Si9B13 amorphous ribbons was 5-10 times faster than other Fe-based catalysts, which is marked with red underlines.

Point 2: Please discuss the relationship between the formation of the crystal phase of Y3 compared with Y6 and Y9 and different Al and Y contents.

Response 2: During the preparation of Y3, Y6 and Y9 ribbons, with the increase of Y element, the formation of α-Al crystal phase and α-Al cluster phase gradually decrease. When the α-Al content decreases, the content of intermetallic compounds will increase.

Point 3: The authors should give references to the melting peak in DRS curves.

Response 3: As for the melting peak in the DSC curves, I have given relevant references [49] in the manuscript.

Ref [49]: Andilab, B.; Ravindran, C.; Dogan, N.; Lombardi, A.; Byczynski, G., In-situ analysis of incipient melting of Al2Cu in a novel high strength Al-Cu casting alloy using laser scanning confocal microscopy. Mater. Charact. 2020, 159, 110064.

Point 4: The authors should explain why the concentration of the MO solution remained almost unchanged in the first 5 min of the reaction.

Response 4: The concentration of MO solution basically remained unchanged five minutes before the reaction because the oxide film on the surface of the ribbon prevented the internal metal elements from participating in the reaction. After five minutes, the oxide film gradually dissolved, and the degradation reaction gradually began, we explain this in detail in section 3.6 of the manuscript.

Point 5: The authors mention that the [H] plays a dominant role in the degradation of MO solution. Thus, the authors should provide some evidence to prove this statement. However, it is not consistent with the explanation of the effect of H2O2 dosage which describes the importance of hydroxyl radical from H2O2.

Response 5: Taking Fig 3a as an example, when H2O2 = 0 mM, the degradation efficiency of Y3 ribbon on MO solution is 70.6%; when H2O2 = 1 mM, the •OH groups can be generated in the solution, which makes the degradation efficiency of Y3 ribbon on MO solution increase to 88.1%, and the degradation efficiency increases by 17.5%. As the degradation efficiency of [H] for MO solution is 70.6%, we say that [H] plays a dominant role in the degradation of MO solution.

Point 6: The authors should test the reusability for the MO degradation over the catalyst.

Response 6: We tested the reusability of the catalyst and provided additional explanations in section 3.6 of the manuscript.

3.6. Reusability of Y3 in degradation MO solution

In order to further study the reusability of the ribbons, we selected Y3 ribbon to repeatedly degrade MO solution under the condition of T = 298 K, pH = 1,  = 1 mM and CMO = 10 mg L-1, as shown in Figure 6. The degradation efficiency η of the Y3 ribbon on MO solution decreased slowly from the 1st to 3rd, and decreased greatly in the 4th. Compared with the 4th, the η of the 5th remained basically unchanged, and still has relatively high degradation efficiency. The normalized concentration Ct/C0 of the MO solution reacting of Y3 ribbon stay almost unchanged during the first 5 min, this may be because in the 1st test, it takes some time to consume the oxide on the surface of the ribbon, and the fresh metal element surface reacts, so the degradation reaction can be achieved quickly in the 2nd to 5th tests. In addition, we found that the degradation rate k of the 2nd was higher than that of the 1st test, which was also related to the elimination of oxide film.

Figure 6. (a) The normalized Ct/C0 change of MO solution during the degradation process of Al88Ni9Y3 (Y3) ribbon from the 1st to the 5th, (b) the degradation efficiency (η = (1 - Ct/C0 × 100%, tr = 45 min) of the degradation process vs. reaction cycles for Y3 ribbon.

Round 2

Reviewer 4 Report

The author revised the manuscript following the suggestion. Thus I would like to accept this manuscript in the present form.